# Evaluating the Erosive Effects of Freshly Squeezed Local Fruit Juices on Human Dental Enamel and Consumption Patterns Among Malaysian Adults

**DOI:** 10.3390/nu17162576

**Published:** 2025-08-08

**Authors:** Zahirrah Begam Mohamed Rasheed, Ahmad Shuhud Irfani Zakaria, Fairuz Abdul Rahman, Erfa Zainialdin, Hazreen Elliana Radzali, Norhafiza Mokhtar, Nurhayati Abdullah, Zaleha Shafiei, Zamirah Zainal Abidin, Mariati Abdul Rahman

**Affiliations:** 1Department of Craniofacial Diagnostics and Bioscience, Faculty of Dentistry, Universiti Kebangsaan Malaysia, Kuala Lumpur 50300, Malaysia; zahirrah.rasheed@ukm.edu.my (Z.B.M.R.); zalehashafiei@ukm.edu.my (Z.S.); zamirah@ukm.edu.my (Z.Z.A.); 2Department of Family Oral Health, Faculty of Dentistry, Universiti Kebangsaan Malaysia, Kuala Lumpur 50300, Malaysia; shuhud_zakaria@ukm.edu.my; 3Stomatology Unit, Institute for Medical Research (IMR), National Institute of Health (NIH), Shah Alam 40170, Selangor, Malaysia; drfairuzrahman@moh.gov.my; 4Klinik Pergigian Erfa, Dataran Wangsa Melawati, Wangsa Melawati, Kuala Lumpur 53300, Malaysia; drerfa@gmail.com; 5Klinik Pergigian I-Smile, Pusat Bandar Senawang, Seremban 70450, Negeri Sembilan, Malaysia; reenell511@yahoo.com; 6Klinik Pergigian Menglembu, Ipoh 31450, Perak, Malaysia; dr.hafizamokhtar@moh.gov.my; 7Pejabat Kesihatan Pergigian, Daerah Padang Terap, Kuala Nerang 06300, Kedah, Malaysia; drnurhayati_abdullah@moh.gov.my

**Keywords:** erosive effect, tooth erosion, acidic fruit juice, microhardness, roughness

## Abstract

Background: The increasing popularity of fruit juices as part of perceived healthy dietary choices has raised concerns regarding their erosive effects on dental enamel. While prior *in vitro* studies have largely relied on commercial fruit drinks and non-human enamel samples, this study adopts a more ecologically valid approach by using fresh local fruit juices and extracted human teeth to evaluate enamel erosion. Objectives: This study aimed to assess the consumption patterns, oral hygiene behaviours, and awareness of the erosive potential of fruit juices among Malaysian adults and to evaluate the erosive effects of freshly squeezed local fruit juices on human dental enamel under simulated oral conditions. Methods: A questionnaire-based cross-sectional survey (*n* = 189) was conducted among dental clinic attendees to assess fruit juice intake habits, oral health practices, and awareness levels. In parallel, an *in vitro* study was performed using 40 extracted premolar teeth immersed in lime juice, pineapple juice, citric acid (positive control), or distilled water (negative control) over a 10-day period. Enamel volume loss, surface roughness, and microhardness were analysed pre- and post-immersion. Results: Fruit juice consumption was highly prevalent, with lime (57.7%) being the most commonly consumed, followed by watermelon (53.0%), star fruit (15.9%), and pineapple (15.4%). The majority of respondents preferred sweetened juices (75.7%) and demonstrated only moderate oral hygiene, with just 53.4% reporting brushing their teeth twice daily. Awareness of the dental effects of acidic beverages was limited. *In vitro* results confirmed that both lime and pineapple juices significantly reduced enamel microhardness and increased surface roughness (*p* < 0.0001), with lime juice causing the greatest enamel volume loss due to its higher acidity. Conclusions: These findings highlight the need for public health strategies that raise awareness on the implications of dietary acids and promote protective oral health behaviours. Dental practitioners should incorporate dietary counselling in routine care, particularly for populations at higher risk.

## 1. Introduction

Tooth wear is an increasingly common dental condition characterised by the progressive loss of hard tooth structure. It encompasses four primary processes namely erosion, attrition, abrasion, and abfraction [1]. Among these, dental erosion is the most prominent and well-documented contributor to tooth wear and is often implicated as a significant reason for patients to seek restorative dental treatment [2].

Dental erosion refers to the irreversible and typically painless loss of enamel and dentine due to acidic dissolution without the involvement of bacterial activity [3]. The primary aetiologic factors are acid of intrinsic factors such as acid reflux or extrinsic factors such as frequent consumption of acidic foods and beverages, use of certain medications, and environmental exposure to acidic agents [4,5]. Although dietary acids are common contributors to dental erosion, the extent of enamel loss is influenced by several modifying factors, including pH [6], titratable acidity [7], buffering capacity [8], undissociated acid concentration, acid type [9], concentration [10], chelating capacity [11], and the quantity and quality of saliva [12], which plays a crucial role in buffering and remineralisation.

The damaging potential of dental erosion increases when combined with other forms of wear such as attrition or abrasion, resulting in more severe tooth surface loss. As erosion progresses beyond the enamel into the dentine layer, it can lead to tooth hypersensitivity, pulp involvement, and even tooth fracture [13]. Additionally, erosive conditions may compromise the clinical performance and longevity of restorative materials, including glass ionomer cements, resin-modified composites, and conventional resin composites [14].

Numerous clinical and *in vitro* studies have established a clear association between the consumption of acidic beverages such as carbonated soft drinks and fruit juices on enamel erosion. For example, fruit juices such as orange, blackcurrant, lime, and pineapple have been shown to possess high erosive potential due to their low pH and high titratable acidity [15,16]. In recent years, there has also been a noticeable shift toward healthier dietary habits, including increased consumption of fruit juices, often perceived as a nutritious alternative to sugary soft drinks and as part of “detox” regimens [17]. However, despite their perceived health benefits, many fruit juices remain highly acidic and pose a risk to enamel integrity [18]. Given the increasing global life expectancy and dietary habits that favour the frequent intake of acidic beverages, dental erosion is expected to become an even more significant challenge in future dental practice. Understanding the erosive potential of commonly consumed beverages is thus essential for the development of effective preventive strategies and evidence-based patient education.

To date, most *in vitro* studies assessing the erosive effects of fruit juices have utilised commercially available cordials [19], powdered fruit drinks [20], or non-human enamel [20], and often involved short immersion periods [15,21]. This study seeks to address these limitations through a more ecologically valid and context-specific approach. First, a questionnaire-based survey will be conducted to evaluate participants’ dietary habits related to fruit juice consumption, their oral health behaviours, and their knowledge and awareness of the potential erosive effects of these beverages. Second, an *in vitro* analysis will be performed using freshly squeezed local fruit juices and human enamel samples exposed to acidic conditions over a prolonged immersion period. The study design also incorporates cycles of artificial saliva immersion to more closely simulate the natural remineralising environment of the oral cavity. Together, these components aim to generate more representative and clinically relevant data on the erosive potential of local fruit juices, contributing to both scientific understanding and public health recommendations.

## 2. Materials and Methods

### 2.1. Subject Recruitments and Questionnaire

Participants were recruited from patients attending Poliklinik Pelajar Pergigian Universiti Kebangsaan Malaysia (UKM). Convenience sampling was employed, and patients were invited to participate voluntarily after being informed about the purpose and nature of the study. Ethical approval was obtained from the Human Research Ethics Committee, Centre for Research and Instrumentation Management, UKM (reference number JEP-2023-805 on 5 January 2024) and informed consent was obtained from all participants prior to data collection.

A self-administered questionnaire was developed to assess the participants’ habits related to the consumption of local fruit juices. Prior to initiating the study, the questionnaire was completed by a pilot group (*n* = 10) to ensure comprehension and legibility. It was a close-ended questionnaire with some questions permitting the subjects to choose multiple answers. The questionnaire includes sections on demographic information, dietary habit, knowledge and awareness on fruit juice acidity and its effects, and oral hygiene awareness.

### 2.2. Fruit Juices Preparation, pH Testing, and Neutralisable Acidity

Fresh juices were prepared using a juice extractor by blending one whole watermelon, 1 kg of star fruit, three pineapples, and 1 kg of lime. The pH of the fruit juices was determined using a pH meter. For titration purposes, 50 mL of fruit juice was placed in a conical flask. Sodium hydroxide (NaOH) solution with a concentration of 0.1 M was used in the titration process to neutralise the acidity of the fruit juices. In each titration, 2–5 mL of NaOH solution was gradually added to the fruit juices. The rise in pH was continuously monitored and recorded until the pH reached neutrality. Each titration was carried out in triplicates.

### 2.3. In Vitro Teeth Selection and Preparation

A total of 40 extracted premolar teeth were collected from patients attending Klinik Rawatan Utama, Fakulti Pergigian Universiti Kebangsaan Malaysia (UKM). Consent was obtained from each patient prior to tooth collection. The tooth selection criteria were sound, free from caries, defect, hypoplasia, cracks, and any restorations. The premolar tooth was chosen because it is easily collected and is the most common tooth extracted prior to orthodontic treatment. The presence of palatal and buccal cusps, and the proximal ridges make it an important reference point for enamel hardness and roughness scanning purposes.

Each tooth was mounted in a mould upright and Vaseline was applied to the mould surfaces to act as a lubricant for ease of separation. Resin was prepared according to the manufacturer’s instructions and allowed to flow into the mould, covering two-thirds of the tooth, exposing the coronal part. The enamel surface of the teeth was then ground and polished to provide a flat surface for data collection. The teeth were randomly divided into two groups, consisting of 20 teeth each, for microhardness testing using Shimadzu HMV-G (Shimadzu Corporation, L0452 Kyoto, Japan) and surface analysis using a 3D optical surface measurement system (Infinite Focus Real 3D, Alicona Imaging GmbH, Grambach, Austria). The teeth were further divided into distilled water (negative control), citric acid (positive control), pineapple juice, and lime juice in both groups.

### 2.4. Artificial Saliva and Citric Acid Preparation

Artificial saliva was prepared to simulate washing effects resembling the chemical environment of the oral cavity. The formulation, adapted from McKnight-Hanes and Whitford [22], was designed to mimic, to a certain degree, the composition and physicochemical properties of natural human saliva, particularly in terms of ionic content, pH, and viscosity. The preparation of 1 L of artificial saliva consists of 1000 mL of distilled water, 10 g of sodium carboxy-methyl cellulose, 0.62 g of potassium chloride, 0.87 g of sodium chloride, 0.06 g of magnesium chloride, 0.17 g of calcium chloride, 0.8 g of di-potassium hydrogen orthophosphate, 0.3 g of potassium di-hydrogen orthophosphate, 0.0044 g of sodium fluoride, 29.95 g of sorbitol, and 1.0 g of methyl p-hydroxybenzoate. The pH was then adjusted to 7.2. Each tooth was immersed in 20 mL of artificial saliva during the intermediate period, post-tooth immersion in fruit juices and citric acid.

Citric acid was prepared by dissolving 0.6 g of citric acid powder in 100 mL of distilled water, yielding a 0.032 M solution. The pH of the citric acid solution was adjusted to 2.6 using 0.1 M sodium hydroxide. Citric acid served as a positive control to compare the erosive effect of the acid with fruit juices.

### 2.5. Teeth Immersion Protocol

The teeth in the negative control group were immersed in distilled water for 15 min three times daily for 10 consecutive days at room temperature. The teeth were kept in a plastic container during the immersion and were submerged in artificial saliva between immersions at room temperature.

The teeth in the positive control group were immersed in 0.032 M citric acid (pH 2.6) for 15 min three times daily for 10 consecutive days at room temperature. The teeth were kept in a plastic container during the immersion and were submerged in artificial saliva between immersions at room temperature.

Teeth in the fruit juice groups were immersed in the respective juice for 15 min three times daily for 10 consecutive days at room temperature. The teeth were kept in a plastic container during the immersion and were submerged in artificial saliva between immersions at room temperature.

### 2.6. Enamel Surface Microhardness Analysis

The Vickers hardness tester (Shimadzu HMV-2000, Kyoto, Japan) was used to evaluate the hardness of the enamel surface post 10 days of tooth immersion in distilled water, citric acid, pineapple juice, and lime juice. The diamond tip of known dimension was pressed onto the flattened enamel surface with a 100 g load for 15 s. Vickers Hardness Number (VHN) was used to record triplicate readings for each tooth. The mean and standard deviation of each reading were calculated using SPSS version 15.0.

### 2.7. Enamel Surface Roughness Analysis

The surface roughness of the enamel was evaluated using a 3D optical surface measurement system (Infinite Focus Real 3D, Alicona Imaging GmbH, Grambach, Austria) after 10 days of tooth immersion in distilled water, citric acid, pineapple juice, and lime juice. A magnification of 20× with a vertical resolution of 219.00 nm and a lateral resolution of 2.94 μm was used to measure the mean surface roughness (Ra) at four different spots: palatal cusp, buccal cusps, mesial, and distal ridge, of each tooth.

### 2.8. Data Analysis

Questionnaire data were analysed using SPSS version 15.0. The mean and standard deviation of pH measurement, enamel volume loss, mean microhardness value, and mean surface roughness were calculated using Graph Pad Prism version 10. Statistical analysis was performed using two-way analysis of variance (ANOVA) followed by Tukey’s post hoc test for multiple comparisons. A *p*-value of <0.05 was considered statistically significant.

## 3. Results

### 3.1. Survey Findings on Fruit Juice Consumption and Oral Health Behaviour

A total of 189 respondents completed the study questionnaire, ranging in age from 13 to 62 years, with a mean age of 27.2 years. The majority of participants were female (*n* = 143) and predominantly of Malay ethnicity (*n* = 161) (Table 1).

Of the 189 respondents surveyed, nearly all reported a preference for consuming fruit juices, with 79.4% indicating that they typically obtained these beverages from restaurants. Despite this high preference, only 14.8% reported frequent consumption, while 52.9% consumed fruit juices occasionally. The majority (74.1%) limited their intake to once per day. The primary motivations for fruit juice consumption included their status as a favourite beverage (65%) and their perceived role as a dietary supplement (42.3%), particularly as a substitute for fresh fruit. The most commonly consumed fruit juices were lime (57.7%), watermelon (53%), star fruit (15.9%), and pineapple (15.4%). A significant proportion of respondents (75.7%) preferred their fruit juices sweetened with added sugar. Other commonly consumed beverages included iced lemon tea (34.3%) and carbonated mixed drinks such as fruit cocktails (8.5%). Fruit juice consumption was most frequent during dinner (56%), followed by lunch (47%). In terms of drinking methods, the majority of participants reported using a straw (78.8%) or sipping directly (61.4%). Post-consumption oral hygiene practices were minimal, with 50% of respondents indicating that they took no action after drinking fruit juice. Only 21.7% reported rinsing with tap water, 6.9% drank plain water, and a mere 4.2% brushed their teeth (Table 2).

Regarding oral health, 53.4% of respondents reported brushing their teeth twice daily, with the use of a toothbrush being the most common method (96.8%), followed by dental floss (33.3%) and mouth rinse (28%). The majority of participants brushed their teeth after waking up (97.9%) and before bedtime (83.1%), whereas only 25.9% brushed after every meal and 10.6% after snacking. Awareness of dental problems was primarily triggered by the presence of pain or symptoms (86.8%) or during dental visits (65.6%). Over three-quarters of respondents (78.8%) sought dental treatment when experiencing issues, while 46.6% opted for self-medication using analgesics, such as paracetamol. Most respondents stated that they visit the dentist only when necessary (56.6%), while only 6.9% of the respondents attend regular monthly check-ups (Table 3).

Respondents’ knowledge and awareness regarding the acidic content of fruit juices, their potential effects, and strategies for prevention or symptom relief were evaluated. A majority of participants were aware that pineapple (82.0%), lime (89.9%), and orange (88.4%) juices are acidic in nature. However, only 69.8% recognised that these juices could lead to gastrointestinal and oral health issues, such as stomach ache, peptic ulcers (58.7%), tooth sensitivity (45.0%), dental caries (30.2%), and tooth staining (11.1%). The relatively low awareness of oral complications was attributed to the absence of noticeable tooth sensitivity during consumption. Nearly half of the respondents (48.1%) reported experiencing stomach discomfort and 39.7% reported tingling sensations on the tongue when consuming pineapple, lime, or both. Additional reported symptoms included tooth sensitivity (22.8%), toothache (13.8%), and erosion of the oral mucosa (14.8%). In response to these symptoms, respondents employed several coping strategies such as rinsing the mouth with tap water (50.8%), discontinuing juice consumption (39.1%), adding ice cubes to the juice (36.0%), brushing teeth after consumption (29.2%), and freezing the juice prior to consumption (26.5%) (Table 4).

### 3.2. In Vitro Evaluation of Enamel Erosion

Subsequently, the selected fruit juices were prepared using fresh fruits commonly consumed by the respondents. Among the tested beverages, lime juice exhibited the highest acidity with a mean pH of 2.43 ± 0.38, followed by star fruit juice (3.79 ± 0.20), pineapple juice (4.12 ± 0.34), and watermelon juice (5.81 ± 0.57). Interestingly, despite its relatively higher pH, pineapple juice required the greatest volume of 0.1 M NaOH solution to achieve neutralisation, with an average of 58.0 ± 2.65 mL. This was followed by lime juice (53.0 ± 5.00 mL), star fruit juice (19.0 ± 1.70 mL), and watermelon juice (15.0 ± 6.40 mL), indicating the influence of titratable acidity in addition to pH in determining erosive potential (Table 5).

Although star fruit exhibited a lower pH compared to pineapple juice, its significantly lower neutralisable acidity suggests a reduced erosive potential. Therefore, it was excluded from the *in vitro* erosion model to prioritise testing beverages with higher acid challenges of lime and pineapple juice. The erosive potential of these acidic fruit juices was evaluated using human tooth enamel, with citric acid serving as the acidic control. Fresh batches of lime and pineapple juices were prepared, and their pH values were measured prior to enamel immersion. In agreement with previous findings, lime juice exhibited the lowest pH (2.91), indicating higher acidity compared to pineapple juice (pH 3.97) (Figure 1A). All enamel specimens demonstrated significant volumetric loss following exposure. Notably, the greatest average volume loss occurred in the lime juice group, consistent with its higher acidity (Figure 1B). Post-immersion assessments revealed a significant reduction in enamel microhardness (Figure 1C) and a marked increase in surface roughness (Figure 1D) for both juices when compared to baseline values (*p* < 0.001). Comparison after 10-day enamel immersion in the solutions to the distilled water also showed a statistically significant reduction in microhardness with citric acid, pineapple, and lime juices (*p* < 0.01) (Figure 2A) while surface roughness showed statistically significant increase with citric acid (*p* < 0.05) and a trend of increase with pineapple and lime juices (Figure 2B). These findings underscore the pronounced demineralising effect of acidic fruit juices on enamel integrity and the greater erosive potential of lime juice relative to pineapple juice, which is consistent with their pH and neutralisable acidity profiles.

## 4. Discussion

The Malaysian fruit juice market is experiencing steady growth and is projected to expand significantly by 2030, driven largely by shifting consumer preferences towards healthier and more natural beverage options [23,24]. In the present study, a substantial proportion of respondents reported regularly consuming fruit juices, with a preference for sourcing these beverages from restaurants. This trend is consistent with national data showing that the frequency of dining out among Malaysians has nearly doubled over the past three decades [25,26], with more than half of the population reportedly eating out at least once a day [27]. This dining pattern likely contributes to the observed daily consumption of fruit juices among respondents in this study.

Taste preference and the perception of fruit juice as a dietary supplement were identified as the primary motivations for consumption. This aligns with previous findings indicating that sensory appeal, particularly taste, is a significant determinant of fruit juice intake among Malaysian adults, with orange, mango, and watermelon being the most commonly preferred varieties [28]. Notably, despite consuming natural fruit juices, over half of the respondents in this study reported adding sugar to their drinks, consistent with national trends of a high weekly intake of sugar-sweetened beverages [29,30]. Watermelon juice, in particular, is popular due to its high-water content and refreshing qualities. Its consumption during lunch or dinner may be influenced by Malaysia’s tropical climate, suggesting its role as a cooling beverage and a complement to meals.

From a cultural and commercial perspective, fruit juices in Malaysia are commonly served in glasses or packaged containers and are typically consumed using straws or sipped, particularly in cafes and restaurants. These patterns were also evident among the respondents in the present study. Although the use of straws has been proposed as a potential method to minimise dental erosion, current evidence highlights that their protective effect is highly dependent on straw positioning and drinking technique. Directing the stream of water toward the teeth may increase the risk of localised erosive wear. Therefore, one way to avoid acid contact with the teeth is to place the straw tip at the back of the mouth, thus allowing the acidic beverage to bypass the teeth and flow directly over the tongue [31]. Similarly, sipping behaviours often promote prolonged oral exposure to acidic beverages due to habits such as ‘holding’ or ‘swishing’ the drink in the mouth, which further exacerbates the risk of enamel erosion [32]. These findings highlight an important behavioural concern that despite the frequent consumption of fruit juices, often with added sugar, a substantial proportion of respondents reported inadequate oral hygiene practices following intake. This combination of high acidic and sugar exposure with suboptimal oral care habits poses a significant threat to long-term oral health [33].

While most respondents adhered to the basic oral hygiene recommendation of brushing twice daily, the use of supplementary oral hygiene aids such as dental floss and mouth rinse was notably less common. Given the erosive nature of fruit juices, especially when combined with added sugars, practices such as rinsing and flossing are crucial for maintaining optimal oral health [34]. Of concern is the finding that a considerable proportion of participants did not perform any oral hygiene activity immediately after consuming fruit juice. This is clinically relevant, as post-consumption behaviour can significantly influence the extent of enamel erosion. Previous studies have highlighted the importance of rinsing with water or using fluoride-containing products after consuming acidic beverages to mitigate their erosive effects [35,36]. Additionally, although many respondents self-reported existing dental issues, few had sought professional dental care, suggesting a gap in preventive behaviour. Timely dental visits are crucial for addressing erosion-related symptoms before they progress into more severe oral health issues [37].

This study also revealed a moderate level of awareness among the respondents regarding the acidic nature of certain fruit juices and their potential health implications such as stomach discomfort and tooth sensitivity. High awareness on the acidic nature of pineapple, lime, and orange by the respondents could be due to the sour taste of the juices, since most respondents depend on the taste sensory in choosing their favourite juices. The presence of the sourness is contributed by the citric acid content that are usually higher in citrus fruits [38]. Although respondents are aware of the acidic content, they lack knowledge about its effects on oral health, such as the development of caries, tooth sensitivity, and tooth staining. Citric acid contributes to enamel erosion by dissolving its mineral content, which weakens the enamel and accelerates carious lesion formation [39]. The progressive loss can lead to deeper penetration into dentin, resulting in tooth sensitivity [40].

To further demonstrate the erosive potential of selected fruit juices, an *in vitro* experiment was conducted using extracted human teeth. The finding demonstrated that exposure to lime and pineapple juices, as well as citric acid, resulted in a measurable decrease in enamel hardness and increased surface roughness. Lime juice was found to be more erosive than pineapple juice, with its effects closely resembling those of citric acid. These findings highlight the potentially harmful effects of regularly consuming lime juice, particularly in the absence of good oral hygiene practices, as it can cause dental erosion and may contribute to long-term deterioration of oral health. Citric acid was used as a positive control as it is a known erosive solution in demineralisation studies [41,42,43]. The immersion protocol, which simulated four daily meals with 15-min exposures, was designed to reflect typical sipping behaviour during meals. Both pineapple and lime juices caused significant reductions in enamel hardness and roughness, supporting existing literature on the erosive capacity of acidic fruit juices [44]. It is important to note that most prior *in vitro* studies have employed commercially available fruit juice cordials [19], powdered drinks [20], or even non-human enamel [20] with short immersion periods [15,21] that do not accurately reflect oral conditions. In contrast, this study utilised freshly squeezed local fruit juices, human enamel specimens, and extended immersion durations, along with alternating cycles of artificial saliva to better replicate the natural remineralising action of saliva. These modifications aimed to enhance the ecological validity of the study and provide more representative insights into the actual effects of local fruit juice consumption on dental enamel.

## 5. Limitations and Future Research

The cross-sectional design precludes the establishment of causal relationship. Additionally, self-reported data on dietary habits and oral hygiene practices may be subjected to recall bias by the respondents. Future longitudinal studies with larger and more diverse populations are warranted to elucidate the relationship between fruit juices consumption and enamel erosion. In the *in vitro* component, the use of premolar teeth may not fully represent the broader structural and compositional variations across the different types of teeth, and the immersion protocol lack of replicating the in vivo condition. Therefore, future studies should consider comparing the effects of freshly squeezed fruit juices on various types of tooth enamel to determine the erosive impacts and adding additional parameters such as temperature, salivary flow, and mastication.

## 6. Conclusions

This study offers important insights into the consumption patterns, oral health behaviours, and awareness levels related to fruit juice intake among Malaysian adults with *in vitro* evidence of the erosive potential of locally consumed fruit juices. The survey showed the habit of frequent intake of acidic beverages with added sugar combined with inadequate post-consumption oral hygiene and limited awareness of dietary acid effects which may increase the risk of dental erosion. The *in vitro* findings confirmed the erosive potential of freshly squeezed pineapple and lime juices, reflected by reduced enamel hardness and increased surface roughness under simulated oral conditions. These highlights the potential value of culturally relevant public health initiatives aiming at increasing the awareness of the risks associated with frequent acidic beverage consumption and encouraging protective oral hygiene behaviours. Dental professionals could consider incorporating dietary counselling into routine care, particularly for individuals at higher risk of dental erosion. Future longitudinal research is needed to confirm these associations and to evaluate the effectiveness of targeted preventive strategies.

## Figures and Tables

**Figure 1 nutrients-17-02576-f001:**
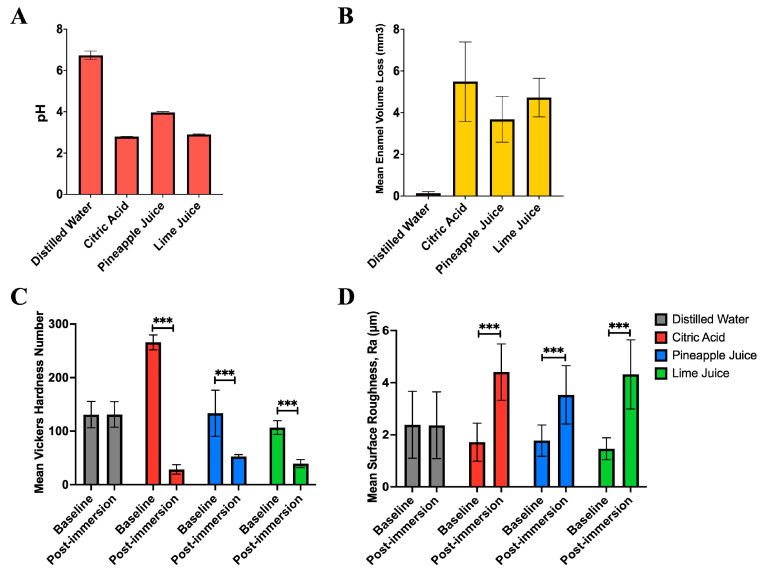
(**A**) Mean pH values of the test solutions. Citric acid showed the lowest pH, followed by lime juice and pineapple juice. (**B**) Mean enamel volume loss (mm^3^) following immersion. Citric acid caused the highest enamel volume loss, followed by lime juice and pineapple juice. (**C**) Mean Vickers Hardness Number (VHN) of enamel specimens post-immersion, indicating significant reductions in hardness for the citric acid, lime, and pineapple juice groups compared to baseline. (**D**) Mean surface roughness (Ra, µm) post-immersion, showing significant increases for all acidic solutions compared to baseline. Significance was determined using two-way ANOVA followed by Tukey’s post hoc test. Asterisks indicate statistically significant difference, *p* < 0.001 (***).

**Figure 2 nutrients-17-02576-f002:**
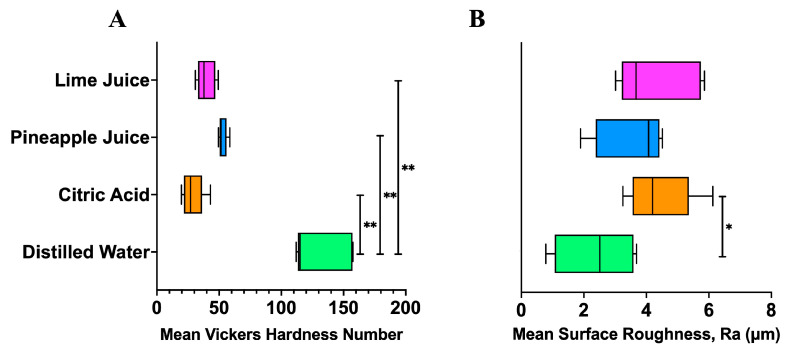
Comparison of enamel microhardness and surface roughness after immersion in test solutions. (**A**) Mean Vickers Hardness Number (VHN) and (**B**) mean surface roughness (Ra, µm) of enamel specimens following 10-day immersion in distilled water (DW), citric acid, pineapple juice, and lime juice. In (**A**), enamel exposed to citric acid, pineapple juice, and lime juice showed significantly lower microhardness compared to DW. In (**B**), only citric acid resulted in a statistically significant increase in surface roughness compared to DW. Bars represent post-immersion mean values ± standard deviation. Statistical significance was determined using two-way ANOVA followed by Tukey’s post hoc test. Asterisks indicate significant differences vs. distilled water: *p* < 0.01 (**) and *p* < 0.05 (*).

**Table 1 nutrients-17-02576-t001:** Respondents’ distribution according to gender and ethnicity.

		Ethnicity				
		Malay [% (No.)]	Chinese [% (No.)]	Indian [% (No.)]	Others [% (No.)]	Total [% (No.)]
**Gender**	Male	19.6 (37)	2.1 (4)	2.1 (4)	0.5 (1)	24.3 (46)
	Female	65.6 (124)	3.2 (6)	2.6 (5)	4.2 (8)	75.7 (143)
**Total**		85.2 (161)	5.3 (10)	4.7 (9)	4.7 (9)	100 (189)

**Table 2 nutrients-17-02576-t002:** Respondents’ dietary habit on fruit juices.

Questions	% of Subjects (No. of Respondents)	
Fruit juice preferences		
Yes	91 (172)	
No	9 (17)	
Sources of fruit juices	Yes	No
Home made	49.2 (93)	50.8 (96)
Drink at restaurant	79.4 (150)	20.6 (39)
Bought from shopping complex	62.4 (118)	37.6 (71)
Carbonated fruit juice	19.6 (37)	80.4 (152)
Cordial fruit juice	50.8 (96)	49.2 (93)
Frequency of drinking fruit juices		
Often	14.8 (28)	
Regularly	13.2 (25)	
Occasionally	52.9 (100)	
Seldom	14.3 (27)	
Never	4.8 (9)	
Frequency of drinking fruit juices in a day		
Once	74.1 (140)	
Twice	7.9 (15)	
Three times	2.1 (4)	
More than three times	1.6 (3)	
Never	14.2 (27)	
Reasons for drinking fruit juices	Yes	No
Favourite drinks	65 (123)	35 (68)
Influence from family and friends	27 (51)	73 (138
As diet supplement	42.3 (80)	57.7 (109)
Part of diet programme	23.3 (44)	76.7 (145)
Enhance appetite	32.8 (62)	67.2 (127)
Types of fruit juices	Yes	No
Watermelon	53 (100)	47 (89)
Star fruit	15.9 (30)	84.1 (159)
Pineapple	15.4 (29)	84.6 (160)
Lime	57.7 (109)	42.3 (80)
Others	63.5 (120)	36.5 (69)
Preparation of fruit juices	Yes	No
With sugar	75.7 (143)	24.3 (46)
No sugar	36 (68)	64 (121)
With milk	29.1 (55)	70.9 (134)
With other drinks (e.g., iced lemon tea)	34.4 (65)	65.6 (124)
With carbonated drinks (e.g., fruit cocktail)	8.5 (16)	91.5 (173)
Time of drinking fruit juices	Yes	No
Breakfast	16.4 (31)	83.6 (158)
Lunch	47 (89)	53 (100)
Dinner	56 (106)	44 (83)
Before bedtime	6.9 (13)	93.1 (176)
At other time	42.3 (80)	57.7 (109)
Methods of drinking fruit juices	Yes	No
Sipping	61.4 (116)	38.6 (73)
Using straw	78.8 (149)	21.2 (40)
Gargle and hold it in mouth for a while	8 (15)	92 (174)
Direct from bottle or cans	38.1 (72)	61.9 (117)
Sipping and gargle	6.4 (12)	93.6 (177)
Action taken after drinking fruit juices	Yes	No
Brush teeth straight away	4.2 (8)	95.8 (181)
Gargle with tap water	21.7 (41)	78.3 (148)
Drink plain water	6.9 (117)	38.1 (72)
Chewing gum	2.1 (4)	97.9 (185)
Do nothing	50.3 (95)	49.7 (94)

**Table 3 nutrients-17-02576-t003:** Respondents’ oral health behaviour.

Questions	% of Subjects (No. of Subjects)	
Daily tooth brushing frequency		
Once	6.9 (13)	
Twice	53.4 (101)	
Three times	32.3 (61)	
After every meal	7.4 (14)	
Cleaning tools	Yes	No
Toothbrush	96.8 (183)	3.2 (6)
Tooth floss	33.3 (63)	66.7 (126)
Mouth rinse	28 (53)	72 (136)
Tongue cleaner	12.7 (24)	87.3 (165)
Chewing unsweetened gum	10.1 (19)	89.9 (170)
Timing of brushing teeth	Yes	No
After waking up from sleep	97.9 (185)	2.1 (4)
After every main meal	25.9 (49)	74.1 (140)
Before bedtime	83.1 (157)	16.9 (32)
After snacking	10.6 (20)	89.4 (169)
Before praying	21.7 (41)	78.3 (148)
Methods of knowing if there is a dental problem	Yes	No
From the dentist	65.6 (124)	34.4 (65)
Self-awareness due to pain or other symptoms	86.8 (164)	13.2 (25)
From friends	10.6 (20)	89.4 (169)
From family members	12.2 (23)	87.8 (166)
From mass media	20.6 (39)	79.3 (150)
Action taken when having a dental problem	Yes	No
Seek immediate relief (e.g., paracetamol, pain killer)	46.6 (88)	53.4 (101)
Seek dental treatment	78.8 (149)	21.2 (40)
Self-healed	32.3 (61)	67.7 (128)
Ignore the problem	12.7 (24)	87.3 (165)
Use traditional remedies	7.9 (15)	92.1 (174)
Frequency of dental visits		
Monthly	6.9 (13)	
Yearly	15.3 (29)	
Twice a year	15.9 (30)	
When necessary (e.g., when pain arises)	56.6 (107)	
Never	5.3 (10)	

**Table 4 nutrients-17-02576-t004:** Respondents’ knowledge and awareness of fruit juice.

Questions	% of Subjects (No. of Subjects)		
Some tropical fruit juice contain acid	Yes	No	Do not know
Watermelon	8.5 (16)	52.4 (99)	39.1 (74)
Star fruit	49.2 (93)	17.5 (33)	33.3 (63)
Pineapple	82 (155)	4.8 (9)	13.2 (25)
Lime	89.9 (170)	1.6 (3)	8.5 (16)
Orange	88.4 (167)	1.1 (2)	10.5 (20)
Fruit juices can cause	Yes	No	Do not know
Stomach ache	69.8 (132)	12.7 (24)	17.5 (33)
Peptic ulcer	58.7 (111)	15.3 (29)	26 (49)
Tooth sensitivity	45 (85)	19 (36)	36 (68)
Carious tooth	30.2 (57)	26.5 (50)	43.3 (82)
Staining tooth	11.1 (21)	37 (70)	51.9 (98)
Feeling of tooth sensitivity during drinking fruit juice	Yes	No	Do not know
	9 (17)	85.7 (162)	5.3 (10)
When drinking pineapple or/and lime juice, do you think that this will happen?	Yes	No	Do not know
Stomach ache	48.1 (91)	42.9 (81)	9 (17)
Tooth sensitivity	22.8 (43)	59.2 (112)	18 (34)
Tooth ache	13.8 (26)	66.1 (125)	20.1 (38)
Tingling tongue	39.7 (75)	45 (85)	15.3 (29)
Erosion of oral mucosa	14.8 (28)	61.4 (116)	23.8 (45)

**Table 5 nutrients-17-02576-t005:** Mean pH and neutralisable acidity of fresh fruit juices.

Fruit Juice	Mean pH (±SD)	Neutralisable Acidity (mL of 0.1 M NaOH ± SD)
Pineapple	4.12 (±0.34)	58 (±2.7)
Star fruit	3.79 (±0.20)	19 (±1.7)
Lime	2.43 (±0.38)	53 (±50.0)
Watermelon	5.81 (±0.57)	15 (±6.4)

## Data Availability

The original contributions presented in this study are included in the article. Further inquiries can be directed to the corresponding author.

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
