# Peer review of "Evaluating the Erosive Effects of Freshly Squeezed Local Fruit Juices on Human Dental Enamel and Consumption Patterns Among Malaysian Adults"

_nutrients, 2025, doi:10.3390/nu17162576_

Round 1

Reviewer 1 Report

Comments and Suggestions for Authors

The authors examined certain dietary habits and oral hygiene / health habits and perceptions in relation to fruit juice consumption, through a questionnaire in a Malaysian population. In addition, they examined the tooth erosive effect of certain fruit juices by in vitro experiments.

Concerning the epidemiological (first) part of the findings are of certain but local interest for the health authorities and providers of Malaysia and perhaps other countries in that geographical region. As a study based on questionnaires, it has certain limitations, some of them being acknowledged by the authors.

Concerning the in vitro experimental part of the study, the results were expected and are not new in a way because the relation of the titratable acidity of a solution and its tooth demineralising effect is well known. These findings are also not clearly related to the epidemiological part of the study. This second part could be shortened and the paper be focused on the epidemiological issue (with the appropriate modification in the title and the text).

Some minor comments to consider are:

1) The authors used the terms "dentin hypersensitivity" and "tooth sensitivity" to denote a tooth condition that is abnormal in terms of tooth reaction to external stimuli. Although both terms (and many others) have been and are used in the literature, the correct term is tooth (or dental) hypersensitivity. Dentin is sensitive by its nature and it can only become hyposensitive if the dentinal tubules become obliterated. The teeth are physiologically sensitive to thermal stimuli (compare with the vitality test). When too sensitive, i.e. to thermal stimuli of lower intensity than for physiological conditions, then they become hypersensitive. We always refer to the tooth and not a single tooth tissue. 

2) Another misleading term is the one of artificial saliva. Although this term is used in the literature for similar solutions, it is important to realise that nobody has created artificial saliva up to now. It is more correct to call it "washing solution" with concentrations of some inorganic element close to those in human saliva. One may question why the sodium carboxy-methyl cellulose was added (it increases the already high concentration of sodium). It is also necessary to mention if the pH was adjusted to a value because the estimated pH (based on the concentration of phosphates) is 7.5, which is higher than the one of saliva and affects solubility of calcium and magnesium.

3) Line 138: "...to stimulate washing effects...", probably it should be "...to simulate...".

Lines 150-161: Use the term tooth or teeth (preferably the latter) not both of them in the various sentences.

4) Table 1. Indicate that the distribution is given in %(No.)

Author Response

Dear Reviewer,

We would like to thank the reviewers for the thoughtful and constructive comments on our manuscript titled Evaluating the Erosive Effects of Freshly Squeezed Local Fruit Juices on Human Dental Enamel and Consumption Patterns Among Malaysian Adults”. We have carefully considered each comment and made the necessary revisions to enhance the clarity and accuracy of our study. We have also addressed our explanation on certain comments. Below, we provide a detailed point-by-point response to each comment, with references to where changes were made in the revised manuscript. 

Point-by-point Response to Reviewer Comments: 

Comment 1:

The authors examined certain dietary habits and oral hygiene / health habits and perceptions in relation to fruit juice consumption, through a questionnaire in a Malaysian population. In addition, they examined the tooth erosive effect of certain fruit juices by in vitro experiments.

Concerning the epidemiological (first) part of the findings are of certain but local interest for the health authorities and providers of Malaysia and perhaps other countries in that geographical region. As a study based on questionnaires, it has certain limitations, some of them being acknowledged by the authors.

Concerning the in vitro experimental part of the study, the results were expected and are not new in a way because the relation of the titratable acidity of a solution and its tooth demineralising effect is well known. These findings are also not clearly related to the epidemiological part of the study. This second part could be shortened and the paper be focused on the epidemiological issue (with the appropriate modification in the title and the text).

Response:

Thank you for your comments. We agree that the general relationship between acidic beverages and enamel demineralisation is well established in the literature. However, our in vitro component specifically examine the effects of freshly squeezed fruit juices that are commonly consumed in Malaysia under a short term immersion protocol, that are closely reflect the typical drinking habits compared to long term exposure (in hours) as in the current literature. Thus we believe our research provides new knowledge and fill the gap in the current literature. While we also believe that presenting both behavioural and experimental data together strengthens the study by linking local consumption pattern with laboratory evidence.   

Comment 2:

The authors used the terms "dentin hypersensitivity" and "tooth sensitivity" to denote a tooth condition that is abnormal in terms of tooth reaction to external stimuli. Although both terms (and many others) have been and are used in the literature, the correct term is tooth (or dental) hypersensitivity. Dentin is sensitive by its nature and it can only become hyposensitive if the dentinal tubules become obliterated. The teeth are physiologically sensitive to thermal stimuli (compare with the vitality test). When too sensitive, i.e. to thermal stimuli of lower intensity than for physiological conditions, then they become hypersensitive. We always refer to the tooth and not a single tooth tissue. 

Response:

Thank you for your comments. We have amended “dentin hypersensitivity” to “tooth hypersensitivity” (Page 2, Line 63).

Comment 3:

Another misleading term is the one of artificial saliva. Although this term is used in the literature for similar solutions, it is important to realise that nobody has created artificial saliva up to now. It is more correct to call it "washing solution" with concentrations of some inorganic element close to those in human saliva. One may question why the sodium carboxy-methyl cellulose was added (it increases the already high concentration of sodium). It is also necessary to mention if the pH was adjusted to a value because the estimated pH (based on the concentration of phosphates) is 7.5, which is higher than the one of saliva and affects solubility of calcium and magnesium.

Response:

Thank you for your comments. We respectfully disagree with your comment on the misleading use of the artificial saliva. Artificial saliva is a widely accepted term used to describe laboratory-prepared or commercially available solutions that stimulate the properties and composition of natural human saliva. It is also extensively used in the in vitro studies in dental research. We have also amend the manuscript by adding “The pH was then adjusted to 7.2” (Page 4, Line 143) to incorporate the information on the pH of the artificial saliva that was used in the research. This pH is within the normal range of human saliva which is between 6.7 until 7.5.

Comment 4:

Line 138: "...to stimulate washing effects...", probably it should be "...to simulate...".

Lines 150-161: Use the term tooth or teeth (preferably the latter) not both of them in the various sentences.

Response:

Thank you for your comments. We have amend the tooth term to “teeth” in the stated lines.

Comment 5:

Table 1. Indicate that the distribution is given in %(No.)

Response:

Thank you for your comment. We have amended Table 1 by including the %(No.) in the revised manuscript.

Reviewer 2 Report

Comments and Suggestions for Authors

Dear authors,

The manuscript addresses an important and timely public health issue, the potential erosive effects of locally consumed fresh fruit juices on dental enamel. The authors combine a cross-sectional survey with an in vitro experimental model using human teeth, which strengthens ecological validity. However, despite the relevance of the topic and the dual methodology, there are several critical flaws in the design, analysis, and interpretation that need to be addressed.

Major Concerns

Importantly, the manuscript currently lacks reference to or compliance with appropriate reporting guidelines.Since the study includes both a cross-sectional observational component and a laboratory-based in vitro experiment, the authors should adhere to relevant EQUATOR Network guidelines. For the survey component, the STROBE(Strengthening the Reporting of Observational Studies in Epidemiology) guideline should be followed. For the in vitro experimental component, although no specific EQUATOR checklist exists, authors should aim to provide sufficient methodological detail, such as using the CRIS (Checklist for Reporting In-vitro Studies) or refer to reporting best practices in dental research. A completed checklist should be included upon revision.

1. In Vitro Study Design Limitations

  • Restricted juice selection: Although four types of juices (lime, pineapple, star fruit, watermelon) were surveyed, only lime and pineapple were tested in vitro. The exclusion of watermelon and star fruit is inadequately justified, particularly since star fruit had a lower pH than pineapple.

  • Sample size and diversity: Only 40 premolars were used, with no demographic data (e.g., age, sex) of donors provided. Premolars alone may not represent broader enamel characteristics.

  • Lack of temperature and flow simulation: The immersion protocol (3 × 15 min/day for 10 days) does not realistically replicate typical juice consumption, which is transient and includes temperature variation and mechanical forces like chewing.

  • Missing control group with commercial acidic beverages: Including a common commercial soft drink or juice would enhance comparability with existing literature.

2. Survey Design and Representativeness

  • Convenience sampling: Participants were recruited solely from a university dental clinic, which likely introduces bias toward higher awareness or different behavior patterns than the general population.

  • Lack of key variables: Socioeconomic status, education level, smoking, and fluoride exposure were not assessed, limiting interpretation.

  • No link between clinical and behavioral data: Although the study includes both behavioral and in vitro components, the authors do not attempt to correlate individual consumption patterns with reported symptoms or enamel outcomes.

3. Data Presentation and Statistical Issues

  • Surface roughness inconsistencies: Figure 2B suggests that lime and pineapple did not significantly increase surface roughness compared to control, despite the text stating otherwise. Clarification is needed.

  • No power calculation: The authors do not report whether their in vitro sample size had adequate power to detect clinically meaningful differences.

  • Lack of multivariate analyses: The survey data could benefit from logistic regression or similar models to identify predictors of poor oral hygiene or awareness.

4. Overinterpretation of Results

  • The statement that lime juice "may accelerate dental caries" is misleading. The study examines erosion, which is a non-cariogenic form of wear. There is no bacterial involvement assessed, nor is caries incidence measured.

  • Assertions about public health implications are not directly supported by longitudinal or causal evidence.

5. Language and Style

  • There are frequent grammatical issues and awkward phrasings, such as “habit after drinking fruit juice” or “sugarless” instead of “unsweetened”.

Minor Comments

  • Please define abbreviations at first use (e.g., VHN, ANOVA)

  • Consider including SEM images to visually demonstrate enamel surface changes.

Author Response

Dear Reviewer,

We would like to thank the reviewers for the thoughtful and constructive comments on our manuscript titled Evaluating the Erosive Effects of Freshly Squeezed Local Fruit Juices on Human Dental Enamel and Consumption Patterns Among Malaysian Adults”. We have carefully considered each comment and made the necessary revisions to enhance the clarity and accuracy of our study. We have also addressed our explanation on certain comments. Below, we provide a detailed point-by-point response to each comment, with references to where changes were made in the revised manuscript. 

Point-by-point Response to Reviewer Comments: 

Comment 1:

Importantly, the manuscript currently lacks reference to or compliance with appropriate reporting guidelines. Since the study includes both a cross-sectional observational component and a laboratory-based in vitro experiment, the authors should adhere to relevant EQUATOR Network guidelines. For the survey component, the STROBE(Strengthening the Reporting of Observational Studies in Epidemiology) guideline should be followed. For the in vitro experimental component, although no specific EQUATOR checklist exists, authors should aim to provide sufficient methodological detail, such as using the CRIS (Checklist for Reporting In-vitro Studies) or refer to reporting best practices in dental research. A completed checklist should be included upon revision.

Response:

Thank you for your suggestion. The questionnaire that we used in this study was designed to capture only descriptive data on participant’s fruit juice consumption, their daily oral health behaviour, and knowledge and awareness of the acidity nature of the fruit juices. Since no clinical data was gathered from the participants, no association between their behaviours and clinical outcomes such as dental caries were investigated. Therefore, I believed our study does not meet the criteria for observational studies addressed by the STROBE guideline.

Regarding our in vitro component in this study, we also believed that our study does not meet the CRIS guideline. Based on the statement in the CRIS guideline publication below, the guideline focuses on in vitro research on dental materials prior to be extrapolated in the clinical settings. Our in vitro study does not involve dental materials nor the results from our in vitro study to be extrapolated into clinical application. Therefore, we believe that CRIS reporting guideline is not needed for our study.

“In vitro studies provide us with the platform to create, compare and check dental materials prior to their clinical application. In vitro research is an integral part of clinical decision-making as this helps the clinician to understand the physical, mechanical, and biological properties of dental materials and dental hard/soft tissues. In vitro studies thereby forms the major proportion of research that is carried out and published in dentistry. Among the articles submitted in Journal of Conservative Dentistry, 82% are in vitro research. A survey of the articles published in few of the leading journals related to dental materials and Endodontics showed that substantial proportion of in vitro studies (Dental materials 98%, International Endodontic Journal 88%, Journal of Endodontics 65%, and Operative Dentistry 74%). It is needless to say that the relevance of in vitro studies cannot be over-emphasized.” Krithikadatta, J., Gopikrishna, V., & Datta, M. (2014)

  1. In Vitro Study Design Limitations

Comment 2:

Restricted juice selection: Although four types of juices (lime, pineapple, star fruit, watermelon) were surveyed, only lime and pineapple were tested in vitro. The exclusion of watermelon and star fruit is inadequately justified, particularly since star fruit had a lower pH than pineapple.

Response:

Thank you for your question. The reason of omitting star fruit in the in vitro experiment was already justified in the original submitted manuscript in Page 8, Line 245-248.

“Although star fruit exhibited a lower pH compared to pineapple juice, its significantly lower neutralisable acidity suggests a reduced erosive potential. Therefore, it was excluded from the in vitro erosion model to prioritise testing beverages with higher acid challenges of lime and pineapple juice.” (Page 8, Line 245-248)

Comment 3:

Sample size and diversity: Only 40 premolars were used, with no demographic data (e.g., age, sex) of donors provided. Premolars alone may not represent broader enamel characteristics.

Response:

The teeth used in our study were collected from adult patients attending the clinic. However, due to anonymisation during ethical collection, donor age and sex were not recorded. We also have included the limitations of using the premolars teeth in the manuscript in Page 11, Line 371-374.

In the in vitro component, the use of premolars teeth may not fully represent the broader structural and compositional variations across the different types of teeth and the immersion protocol lack of replicating the in vivo condition. Therefore, future studies should consider comparing the effects of freshly squeezed fruit juices on various types of tooth enamel to determine the erosive impacts and adding additional parameters such as temperature, salivary flow, and mastication.” (Page 11, Line 371-376)

Comment 4:

Lack of temperature and flow simulation: The immersion protocol (3 × 15 min/day for 10 days) does not realistically replicate typical juice consumption, which is transient and includes temperature variation and mechanical forces like chewing.

Response:

The immersion protocol used in this study is intentionally designed to be more realistic than many other published protocols in the literature. Most studies often use continuous immersion for several hours, which this approach does not truly reflect typical human drinking behaviour. In contrast, our short repeated exposures better mirrors the duration of juice contact with enamel during normal consumption. Nevertheless, we agree that future studies should incorporate additional physiological parameters such as temperature changes, mechanical forces and salivary flow to better mimic the clinical relevance of the model. This limitation has been added in Page 11, Line 371-374.

In the in vitro component, the use of premolars teeth may not fully represent the broader structural and compositional variations across the different types of teeth and the immersion protocol lack of replicating the in vivo condition. Therefore, future studies should consider comparing the effects of freshly squeezed fruit juices on various types of tooth enamel to determine the erosive impacts and adding additional parameters such as temperature, salivary flow, and mastication.” (Page 11, Line 371-376)

Comment 5:

Missing control group with commercial acidic beverages: Including a common commercial soft drink or juice would enhance comparability with existing literature.

Response:

Thank you for the suggestion. The main aim of this study is to specifically investigate the erosive potential of freshly squeezed fruit juices on enamel. This area is currently underexplored in the literature and we do not have any intention of comparing these beverages with the commercially available acidic drinks in the market, which have been extensively studied and reported. By focusing on freshly squeezed fruit juices erosive effects on enamel, we sought to generate novel data and fill the gap in the literature. We thus believe that our data strengthens the originality and contribution in the erosion field. Other research team could fill the gap of comparing the erosive effects of commercially available fruit juices in the marker versus freshly squeezed fruit juice on enamel.

  1. Survey Design and Representativeness

Comment 6:

Convenience sampling: Participants were recruited solely from a university dental clinic, which likely introduces bias toward higher awareness or different behavior patterns than the general population.

Response:

Thank you for your insight, however we respectfully disagree that our sampling approach introduced a significant bias. Convenience sampling is a common method employed in exploratory studies and is appropriate when the study is exploration rather than generalisation to a broader population. The questionnaire designed in this study is to capture general oral health behaviour and general knowledge and awareness of drinking acidic fruit juices rather than clinical treatment outcomes or condition-specific data. The questions were not directed towards the treatment they were receiving at the clinic nor the responses were analysed in relation to their clinical status. Moreover, university dental clinic patients does not only restricted to the members of the university but are open to the public, thus the pool of participants vary in terms of socioeconomic and educational backgrounds.

Comment 7:

Lack of key variables: Socioeconomic status, education level, smoking, and fluoride exposure were not assessed, limiting interpretation.

Response:

Thank you for your question. The main focus of our questionnaire was to obtain descriptive information on freshly squeezed fruit juice consumption habits, awareness of acidity, and general oral health behaviours of our respondents such as frequency of brushing after drinking fruit juice, the timing, and action taken when experiencing dental problems. We did not collect clinical data such as caries status or any other dental problems data to attempt correlating their behaviour with oral diseases outcome with fruit juices consumption. Therefore, we believe that variables such as smoking habits and fluoride exposure is important in examining the aetiology or risk factors for caries and erosion, while we are not addressing this aims in our study. Similarly, while socioeconomic and education level can influence oral health behaviours, our study was not designed to perform stratified or comparative analyses across different demographic groups. Our goal was to provide an initial descriptive overview of behaviours and awareness in our study population and that is the reason we are focusing on the variables directly relevant to fruit juice consumption habits and oral health practises. While we also agree that future studies could aim at exploring the associations between freshly squeeze fruit juice consumption, awareness, oral health outcomes, demographic and socioeconomic factors.

Comment 8:

No link between clinical and behavioral data: Although the study includes both behavioral and in vitro components, the authors do not attempt to correlate individual consumption patterns with reported symptoms or enamel outcomes.

Response:

Thank you for your question. Our study was not designed to link the questionnaire responses to the in vitro enamel erosion findings at an individual level. The questionnaire responses on the behavioural component aimed to provide a descriptive overview of fruit juice consumption habits while the in vitro component was conducted independently to experimentally assess the erosive potential of selected freshly squeezed fruit juices on the extracted human teeth. There is no intention to establish a causal or correlation relationship between individual consumption patterns and enamel outcomes in our study.

  1. Data Presentation and Statistical Issues

Comment 9:

Surface roughness inconsistencies: Figure 2B suggests that lime and pineapple did not significantly increase surface roughness compared to control, despite the text stating otherwise. Clarification is needed.

Response:

Thank you for your question. In the text body, we stated:

“……while surface roughness showed statistically significant increase with citric acid (p<0.05) and a trend of increase with pineapple and lime juices (Figure 2B).” (Page 9, Line 260-262).

While in the figure caption:

“In (B), only citric acid resulted in a statistically significant increase in surface roughness compared to DW.” (Page 10, Line 279-280).

Comment 10:

No power calculation: The authors do not report whether their in vitro sample size had adequate power to detect clinically meaningful differences.

Response:

Thank you for your question. Power calculation and formal sample size calculations are designed for population-based studies where the aim is to estimate or detect effects within a defined level of confidence. Our study is an in vitro exploratory study with the aim to assess the erosive potential of selected freshly squeezed fruit juices under controlled laboratory conditions (such as using artificial saliva and time-point immersion). Since we do not collect any clinical data and no correlation or relationship analysis were performed, we are not aiming at detecting clinically meaningful differences in our in vitro study. We agree that for confirmatory in vitro studies with any correlation made with clinical data intended to provide definite effect estimates, formal power analysis would be beneficial.

Comment 11:

Lack of multivariate analyses: The survey data could benefit from logistic regression or similar models to identify predictors of poor oral hygiene or awareness.

Response:

We appreciate your suggestion, however, our questionnaire component was designed for descriptive purposes aiming to summarise fruit juice consumption habits, awareness of fruit juice acidity, and general oral health behaviours in our respondents. The data was not structured for a predictive modelling and we do not collect a comprehensive set of predictors variables as you have mentioned in other comments such as socioeconomic status, educational level, fluoride exposure, and smoking for making robust multivariate analysis. This method is valuable in hypothesis-driven analytic studies while our study is exploratory and descriptive and may not benefit from incorporating these analytical approaches.

  1. Overinterpretation of Results

Comment 12:

The statement that lime juice "may accelerate dental caries" is misleading. The study examines erosion, which is a non-cariogenic form of wear. There is no bacterial involvement assessed, nor is caries incidence measured.

Response:

We agree and have removed this statement. The revised statement in the manuscript is as follows:

“These findings highlight the potentially harmful effects of regularly consuming lime juice, particularly in the absence of good oral hygiene practices, as it can cause dental erosion and may contribute to long-term deterioration of oral health.” (Page 11, Line 349-352).

Comment 13:

Assertions about public health implications are not directly supported by longitudinal or causal evidence.

Response:

Thank you for your comment. We have revised the concluding paragraph and we have removed the overstated claims and reworded the conclusion for accuracy.

These highlights the potential value of culturally relevant public health initiatives aiming at increasing the awareness of the risks associated with frequent acidic beverage consumption and encouraging protective oral hygiene behaviours. Dental professionals could consider incorporating dietary counselling into routine care, particularly for individuals at higher risk of dental erosion. Future longitudinal research is needed to confirm these associations and to evaluate the effectiveness of targeted preventive strategies.” (Page 12, Line 385-391).

  1. Language and Style

Comment 14:

There are frequent grammatical issues and awkward phrasings, such as “habit after drinking fruit juice” or “sugarless” instead of “unsweetened”.

Response:

Thank you for your comments. The manuscript has been carefully reviewed and revised for grammar, clarity, and academic tone. Awkward phrases:

“Chewing sugarless gum” has been corrected to “Chewing unsweetened gum” (Page 7, Table 3)

“Habit after drinking fruit juice” has been corrected to “Action taken after drinking fruit juices” (Page 6, Table 2)

“Sugarless” has been corrected to “No sugar” (Page 6, Table 2).

Minor Comments

Comment 15:

Please define abbreviations at first use (e.g., VHN, ANOVA)

Response:

The VHN was already been spelled out in the original submitted manuscript (Page 4, Line 166-167) and ANOVA was also already been spelled out in the original submitted manuscript (Page 4, Line 180).

Comment 16:

Consider including SEM images to visually demonstrate enamel surface changes.

Response:

Thank you for your suggestion. The 3D surface profilometry method that we used in this study are well-established, highly sensitive, and provide robust numerical data on changes in surface roughness, which directly addresses the objectives of our research. While SEM could complement profilometry measurement in the future research, our current data is sufficient to demonstrate the erosive effects of the fruit juices on the enamel surface. We therefore believe that the absence of SEM images do not detract the validity and interpretability of our findings in this manuscript. 

Round 2

Reviewer 1 Report

Comments and Suggestions for Authors

Thank you for revising the article.

Regarding the issue of artificial saliva, it is important to emphasize that scientific validity is not established by the frequency with which a formulation is cited, but rather by the evidence supporting its properties. The article does not provide a scientific rationale for why the solution in question should be considered artificial saliva. Does this formulation truly replicate the key characteristics and functions of natural saliva?

Author Response

Dear Reviewer,

Thank you for your comment. We have amended the "artificial saliva" statement in the method section as follows:

"Artificial saliva was prepared to simulate washing effects resembling the chemical environment of the oral cavity. The formulation, adapted from McKnight-Hanes and Whitford [1], was designed to mimic, to a certain degree, the composition and physicochemical properties of natural human saliva, particularly in terms of ionic content, pH, and viscosity. The preparation of 1L of artificial saliva consists of 1000mL of distilled water, 10g of sodium carboxy-methyl cellulose, 0.62g of potassium chloride, 0.87g of sodium chloride, 0.06g of magnesium chloride, 0.17g of calcium chloride, 0.8g of di-potassium hydrogen orthophosphate, 0.3g of potassium di-hydrogen orthophosphate, 0.0044g of sodium fluoride, 29.95g of sorbitol, and 1.0g of metyl p-hydroxybenzoate. The pH was then adjusted to 7.2. Each tooth was immersed in 20mL of artificial saliva during the intermediate period, post-tooth immersion in the fruit juices and citric acid." (Page 3 and 4, Line 138-148).

Reviewer 2 Report

Comments and Suggestions for Authors

Thnak you for considering my comments

Author Response

Dear Reviewer,

Thank you again for the constructive comments on our manuscript.